# Historical record of *Corallium rubrum* and its changing carbon sequestration capacity: A meta-analysis from the North Western Mediterranean

Miguel Mallo[1]*, Patrizia Ziveri[1,2], Victoria Reyes-García[1,2], Sergio Rossi[1,3]

**1** Institut de Ciència i Tecnologia (ICTA), Universitat Autònoma de Barcelona (UAB), Bellaterra, Barcelona, Spain, **2** Institució Catalana de Recerca i Estudis Avançats (ICREA), Barcelona, Spain, **3** Dipartimento di Scienze e Tecnologie Biologiche e Ambientali (DiSTeBA), Università del Salento, Lecce, Italy

* miguel.mallo@uab.cat

**Data Availability Statement:** All relevant data are within the manuscript and its Supporting Information files.

## Abstract

### Background

There is a scarcity of long time-span and geographically wide research on the health status of *Corallium rubrum*, including limited research on its historical ecology and carbon sequestration capacity.

### Objectives

To reconstruct the temporal trends of the most reported *C. rubrum* population parameters in the Northwestern Mediterranean Sea and to determine the changes in total carbon sequestration by this species.

### Data sources

Quantitative and qualitative, academic and grey documents were collected from scientific web browsers, scientific libraries, and requests to scientists.

### Study eligibility criteria

Documents with original information of basal diameter, height and/or weight per colony, with a depth limit of 60 m in the Catalan and Ligurian Seas were analyzed.

### Synthesis methods

We calculated yearly average values of *C. rubrum* biometric parameters, as well as estimated total weight, carbon flux, and carbon fixation in the structures of *C. rubrum*'s colonies.

### Results

In both study areas, the values of the selected morphometric parameters for *C. rubrum* decreased until the 1990s, then increased from the 2000s, with average values surpassing the levels of the 1960s (Ligurian Sea) or reaching levels slightly lower than those of the

**Funding:** MM has received funding from the Spanish Ministry of Economy and Competitiveness (FPI/MDM-2015-0552-16-3) through the "María de Maeztu" Programme for Units of Excellence. SR was funded by a Marie Curie International Outgoing Fellowship (Animal Forest Health, Grant Agreement Number 327845) and from P-SPHERE (COFUND Marie Curie, Grant Agreement Number 665919). Authors want to thank the support of the Generalitat de Catalunya to MERS (2017 SGR-1588) and CALMED project (CTM2016-79547-R). The funders had no role in study design, data collection and analysis, decision to publish, or preparation of the manuscript.

**Competing interests:** The authors have declared that no competing interests exist.

1980s (Catalan Sea). The difference in carbon sequestered between the oldest (1960s: Ligurian Sea; 1970s: Catalan Sea) and the lowest (1990s) biomass value of colonies is nearly double.

## Limitations

Quantitative data previous to the 1990s are very limited. Information on recent recovery trends in *C. rubrum* parameters is concentrated in a few areas and biased towards colonies in marine protected areas, with scarce quantitative information from colonies in other areas.

## Conclusions

The halt in the *C. rubrum* decreasing trend coincided with the exhaustion of tree-like colonies and the first recovery response due to effective protection measures in some areas. Nevertheless, *C. rubrum* climate change mitigation capacity through carbon sequestration can be drastically reduced from its potential in only a few decades.

## Introduction

The coralligenous is one of the most biodiverse habitats of the Mediterranean Sea [1, 2]. This community forms a complex 3D structure that serves as habitat, protection, and feeding area for several species. Moreover, the community is dominated by benthic suspension feeders [3] that strain suspended matter and food particles and fix the suspended particles into long-lived structures, acting thus as carbon sinks [4]. The coralligenous' structural complexity and the different architecture of the ecosystem-engineering species allow for the development of a large number of highly biodiverse systems that take advantage of the microenvironmental heterogeneity, in which irradiance, water motion, nutrient availability, and other environmental factors highly vary [1, 3]. However, several factors are threatening this habitat, resulting in some suspension feeding communities losing individuals or colonies [5–8]. Moreover, compared to larger specimens, the younger (and smaller) ones possibly retain less carbon in their structures [4, 9]. Biodiversity in this community is also decreasing; the habitat becomes more homogenous and the 3D structures less common [9, 10], and thus less resilient to environmental changes [11, 12].

One of the most emblematic species in the coralligenous community is the red coral, *Corallium rubrum* (CR) [3, 13]. Red coral is a long-living, slow-growing, sciaphilous, and heterotrophic cnidarian that increases the 3D complexity of its habitat. The arborescent structure of red coral may serve as refuge for several organisms [13]. This anthozoan lives in cracks, crevices, overhangs, and boulders between 7 and 1016 m depth [14–16]. A main feature of this passive benthic suspension feeder is its calcium carbonate ($CaCO_3$) skeleton, composed of annual growth bands with an organic skeleton matrix [17]. The red coral captures detrital particulate organic carbon (POC) and nano-, phyto- and zoo-plankton from the water column, from which it obtains carbon and other nutrients. The biomineralization forms an axial skeleton and the sclerites made of $CaCO_3$ crystallize in the form of high-magnesium calcite and organic matter [18]. This process contributes to the storage and biological removal of carbon from seawater [19].

Red coral has been harvested since ancient times, mainly for jewellery [13, 20, 21]. For the jewellery industry, the most valuable pieces are the largest and more branched colonies, which

are also the ones with higher reproductive potential [3, 9, 14, 22]. Reproduction in this organism is limited, as CR is a gonochoric internal brooder with low reproductive rates and limited dispersal of its larvae, resulting in low genetic mixing [23, 24]. Such facts, added to its slow recovery after a perturbation (the recovery process can last from several decades to centuries), has resulted in most of the actual CR colonies being small in size [25–27]. Moreover, in several regions this species is considered 'ecologically extinct' (unable to make its function in the ecosystem) [13, 28], the discovery of large long-lived non-perturbed colonies being nowadays an exception [29, 30]. The extinction of red coral is of concern not only for the species itself, but also for the potential effects on the entire habitat [13].

Very few studies have assessed the historical CR population structure and health status (defined as the normal demographical, physiological and reproductive conditions of a population that would allow perpetuation of future generations, avoiding local extinction [31]) through time. Moreover, the comparison of CR population structure and health status between different areas and large periods of time remains scarcely reported [25, 32–34]. Historical ecology can help trace and interpret the changes between ancient and current populations and ecosystems and the potential drivers of these changes (e.g., biodiversity loss, acceleration of biogeochemical cycles, presence of eutrophic areas, and loss of carbon retention in long-lived structures) [35]. Studies covering wide regions and a large time-span (like those done with other species and ecosystems, e.g., [36–42]) can shed light on CR's temporal trends, which in turn, may help us understand not only the species health status trends, but also the associated loss of ecosystem services, including CR's role as a carbon sink or as a habitat-forming species [13].

In this study, we use historical data on CR biometric and population structure from two different NW Mediterranean regions, Catalan Sea and Ligurian Sea, to quantify CR population changes. The analysis of temporal patterns in CR distribution and biometric parameters is then used to provide an estimate of CR populations as potential carbon sinks.

## Methodology

An extensive literature review was performed regarding the presence, demography, and biometrics of CR in the Catalan Sea (including Côte Vermeille) and the Ligurian Sea regions. Demographic and biometric parameters describe the characteristics of a population and serve to determine the species' health status. Our search included grey and academic literature and documents that offered both quantitative and qualitative descriptions of CR's health status. Specifically, we did a research in the fields "Topic" (Web of Science) and "In the Title" (Google Scholar) using the following keywords: "Corallium rubrum" OR "red coral" OR "coralligenous" AND "Mediterranean" OR "Catalunya" OR "Catalonia" OR "Catalan" OR "Liguria". Additionally, a systematic review of two scientific libraries, plus punctual research in several others, was realized. Additional documents were obtained from our network of scientists working on the topic. Finally, references cited in previously selected documents were also reviewed. Documents were reviewed if written in Catalan, Spanish, English, French, Italian, or Latin. The bibliographic research was not limited regarding the antiquity of a document and we considered documents published up until September 2018. The oldest document found describes CR from the 3$^{rd}$ century BP [43].

From a total of 84 pre-selected documents, 65 contained original data (new scientific knowledge) on CR health status (Fig 1; S1 Table), 22 of them providing only qualitative descriptions. The documents providing quantitative information on the most commonly studied parameters (i.e., basal diameter, height, and dry weight) were reviewed, resulting in 28 documents that included information on at least one of these parameters. Finally, 24 documents

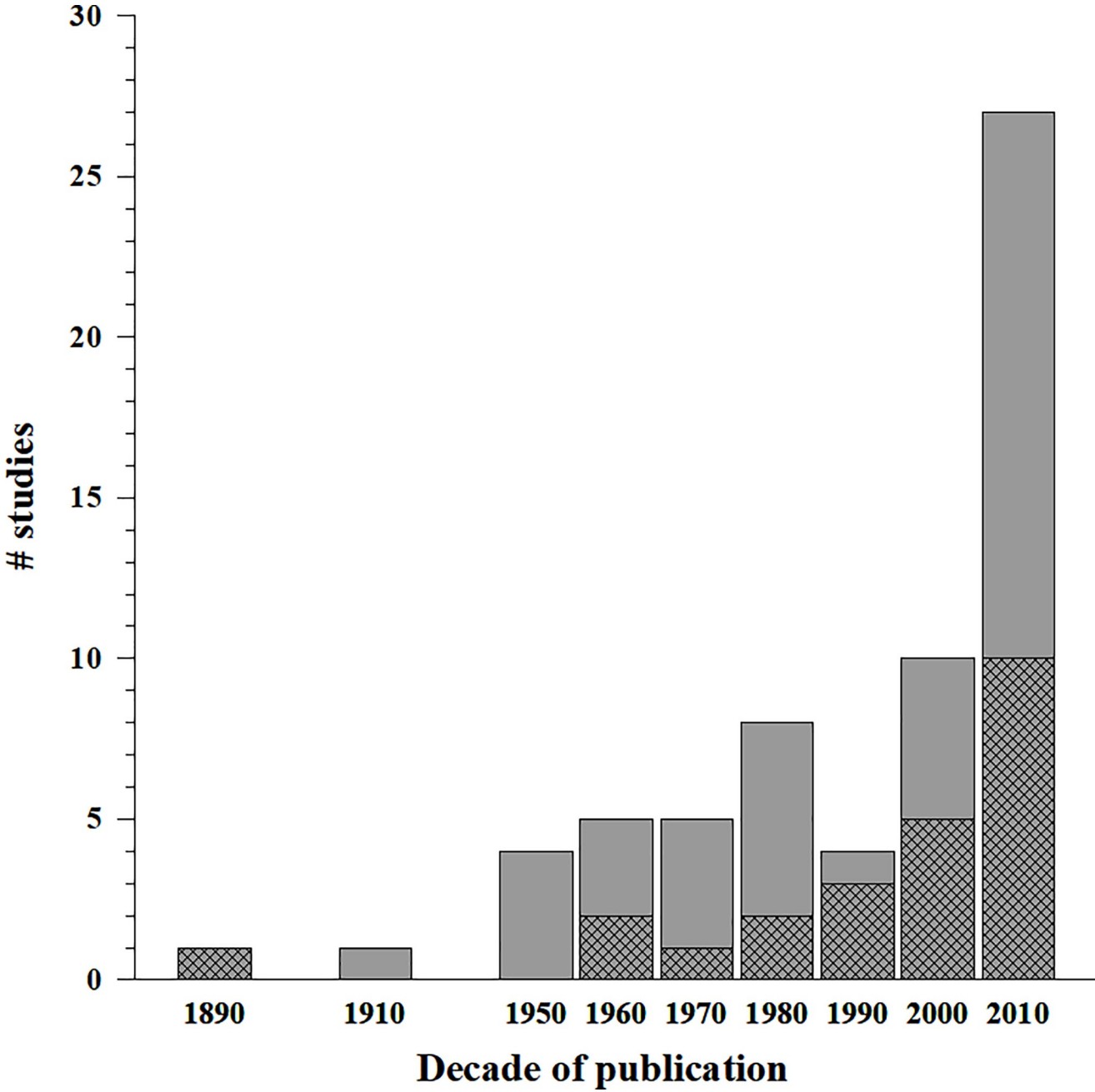

**Fig 1. Publication date and number of documents with original data on *Corallium rubrum* health status, by decade and study site.** In grey the number of documents in the Catalan Sea (including Côte Vermeille) and the Ligurian Sea: n = 65; in cross grids the documents selected for the analysis: n = 24.

with comparable information were included in the meta-analysis (Fig 1). The number of documents selected during the various stages of the meta-analysis is shown in Fig 2.

We used the colony patch as sampling unit for this work. We consider a patch of red coral as a distinct group of colonies, where the distance of colonies within the patch is substantially lower than the distance of colonies between patches (CR usually follows a patchy distribution) [14]. When the colony patch sampling unit was not possible, the nearest higher-resolution available was used. For a given parameter, the yearly mean value was taken as the average of

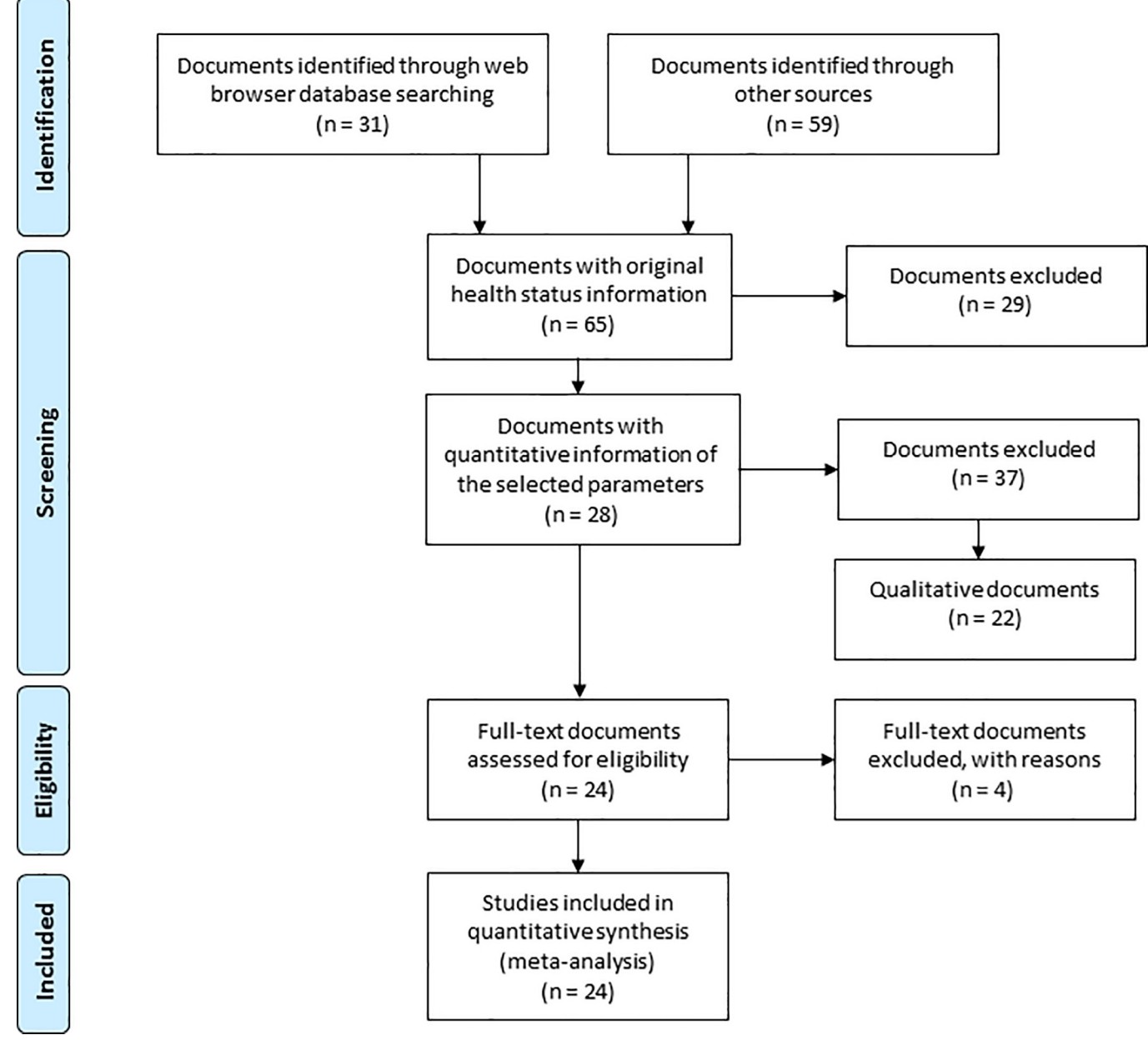

**Fig 2. Selection of documents for inclusion in the meta-analysis.**

the values for all the colony patches (or higher resolution) sampled in the same year. The maximum depth of CR used is 60 m, as the studies made below this depth are very scarce and sparse, decreasing the consistency and precision of statistical comparisons [14, 44–47]. Some documents (25%) did not indicate the sampling year, so we assumed collection was done one year before the publication date. If the parameter was obtained during various years and only an average value was provided, we assigned the result to the last year when samples were reportedly collected (e.g., [15, 48]). Number of colonies used to estimate each parameter can be found in S4 Table.

Results obtained from random sampling were considered comparable, irrespectively of whether data were obtained from quadrants or stations (e.g., [32, 49]), transect lines (e.g., [15, 45]), or covered the whole area (e.g., [48]). Studies excluding colonies < 2 mm basal diameter or < 2 cm height were also considered comparable (e.g., [33]), as it is usually hard to do reliable analysis with such small sizes. Samples obtained through poaching were discarded from the analysis [32, 50–52], since it is unknown whether they represent the natural population accurately. Commercial samples were also discarded as they usually only include the largest individuals (e.g., [44, 53]), and therefore are not representative.

We used all the available data in the literature from our study regions, plus off the Tuscany region, to calculate additional parameters than those found in the literature. The additional values were obtained using regression analysis for each pair of parameters previously selected (Fig 3; See S1 Text, S1 Table, and S1 Fig). The following equations resulted from those regressions:

$$y = 0.875x - 0.655 \qquad (1)$$

y = height (cm), x = basal diameter (mm), $r^2 = 0.651$

$$y = 0.453x + 2.484 \qquad (2)$$

y = height (cm), x = weight (g), $r^2 = 0.552$

$$y = 1.808x - 5.582 \qquad (3)$$

y = weight (g), x = basal diameter (mm), $r^2 = 0.828$

To test the percentage of error of the linear regression Eqs (1–3), we compared the parameter values resulting from our calculations with experimental values obtained from the previous literature research. We used the following formula:

$$\%\text{error} = \left| \frac{\text{calculated value} - \text{literature value}}{\text{literature value}} \right| \times 100 \qquad (4)$$

Formulas (1–3) have 26%, 24% and 47% of error respectively. The data points obtained through formulas represent 40% of the total values used in this study.

To estimate CR biomass, we used the average dry weight values of each year multiplied by a fixed and robust value of density in each region (the data with more replicates and statistically sounder). The density values used are 114.68 colonies/$m^2$ from the Catalan Sea [14] and 216.65 colonies/$m^2$ from the Ligurian Sea [33]. We used a fixed density value as the values obtained from the literature presented a high variability (S2 Fig), presumably from the use of different methodologies and very diverse sample sizes, resulting in incomparable results. The use of a mean fixed density value involves an estimate of the CR biomass for the selected sites, as we do not take into account the spatial and temporal variability of CR's density [30, 32].

The potential area where CR can live was estimated from the literature. We define "potential area" as the surface suitable for CR growth even if is not currently present in the area. In both regions, we limited the extension of the potential areas to the locations where most CR is found. For the Catalan Sea we used data from the three marine protected areas (MPAs): Cap de Creus [54], Medes i Montgrí [55], and Cerbère-Banyuls [56]. For the Ligurian Sea we used the data from Portofino (MPA since 1998) by Cánovas-Molina et al. (2016) [57].

To calculate the potential for carbon sequestration, we obtained the CR carbon ingestion rate of detrital POC and phyto- and zoo-plankton from Tsounis et al. (2006) [58]. The carbon ingestion from bacterioplankton was extracted from Picciano and Ferrier-Pagès (2007) [19]. Oxygen consumption at 16°C was obtained from Previati et al. (2010) [59]. The value was

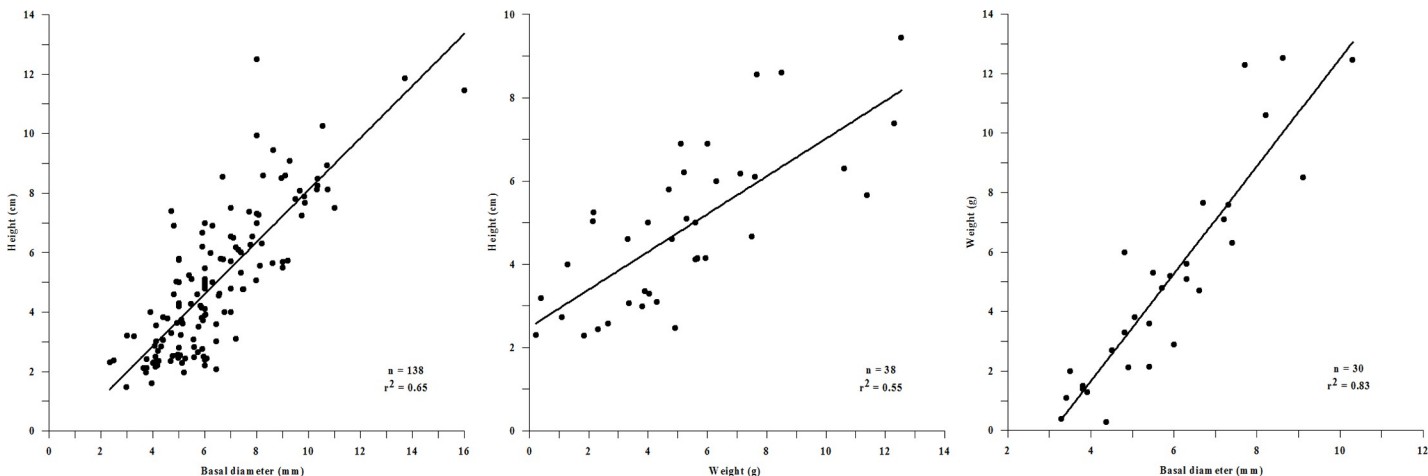

**Fig 3. Scatter plots of *C. rubrum* physiological parameters.** Regression line equations: (1) height vs basal diameter, (2) height vs weight, and (3) weight vs basal diameter. See S1 Text and S1 Table for the data source.

converted into respired carbon equivalents using the conversion factor 0.281 [60]. The estimation of the number of polyps in a colony was done using the results of Santangelo et al. (2003) [22]. The CR's basal diameter growth rate was obtained from Bramanti et al. (2014) [49], using 0.241 and 0.237 mm $y^{-1}$ for the Ligurian and Catalan Sea, respectively. We draw upon Coppari (2015) [61] and use a relation of 5.82 mg $cm^{-1}$ to convert the increase of growth to weight. Like we did for the biometric parameters, we use the same fixed density values, not taking into account the CR's self-thinning processes [30]. To estimate the potential for carbon fixation we only take into account the basal diameter growth. Due to lack of conclusive studies on the topic, we do not consider the increase of total height in the colony, nor the branches. For that reason, our results of carbon retention are expected to be underestimated.

## Results

### Biometric parameters analysis

Twelve documents provide qualitative information about CR's health status for the Catalan Sea: ten documents contain information about CR's abundance, nine about location (geographical and/or bathymetric), two about size, one about diameter, and five about other parameters. For the Ligurian Sea, ten documents providing qualitative information were found. Size was reported in nine documents, location in nine, abundance in eight, weight in two, and other parameters in four (S2 Table).

The qualitative data gathered during the literature review show that CR was exploited since ancient times and from very shallow depths. For example, the illustrations of Tescione (1968) [21] show people harvesting coral in apnea in the 18th century, suggesting that large branches of CR were available at few meters depth in the Gulf of Naples. The text provided by Tescione also shows that, as early as the 14th century there was a real concern about the use of the *ingegno* and the intensive harvesting of the species. The following texts illustrate this concern: "...the fear that coral reefs would soon be exhausted, since, by a royal edict of 1332–33, the prohibition to fish for coral without the King's permission between Cape Minerva and Capri, a place very rich in coral, was renewed."; "To face the threat of impoverishment of the sea-bottoms, because of the devastating action of the devices that were being used, the French tried very hard to find better means of fishing, and the Academy of Marseilles, in 1876, announced a useless competition with a prize for the inventor of a less ravaging device." [21] (S2 Table).

The reviewed literature also suggests that CR was more abundant before the publication of the first studies with occupancy data. This is evident in the following texts: "...fishermen that keep the ones that casually extract with the fish." [62]; "...at the beach, thrown by the waves, especially after storms..." [63]; "...the visions all the coral harvesters remember are about all the rocky walls and caves completely crowded of flowered coral with its white polyps, and from the big and abundant quantity and variety of fishes and lobsters, it is an inexplicable wonder..." [64]; "...rapidly plunging into the water pulled out corals." [21] (S2 Table).

Regarding quantitative data, 29% of the documents included in this study yield data coming from grey literature, 37% from the scientific web browsers search, 46% from the scientific libraries and the network of scientists, and 21% from the references in the pre-selected literature. The most represented region is the Catalan Sea, with 58% of the documents referring to this area. The Ligurian Sea accounts for 50% of the documents. Two documents provide information from both sides (S1 Table). All the studies from the Ligurian Sea were sampled in Portofino, with the exception of the roman writer Gaius Julius Solinus (1895) [43], a document in which the most specific location stated is "Liguria". One third of the samples for the Catalan Sea come from Cap de Creus, another third from Medes Islands and the Montgrí Coast, and almost all the remainder from the Côte Vermeille. One study also includes samples from Begur coast (S1 Table).

For both regions and for all parameters analysed, we find a "U" shaped distribution of the data across time, best fitted by a $2^{nd}$ degree polynomic function (Fig 4, S4 Table). The Catalan Sea region contains more data points per year and more years of data than the Ligurian Sea. In both regions, however, the 1990s mark a clear difference in the amount of data available, with most of the data points belonging to the period 1990–2005. Overall, data for the analysed parameters have increased more in the Ligurian Sea than in the Catalan Sea. The Ligurian Sea also contains older data with several studies conducted during the first half of the 1960s. In contrast the first study for the Catalan Sea was conducted in the late-1970s.

In the Catalan Sea, the values of the parameters observed experienced a decrease until the 1990s. These values remained low until the mid-2000s. After the mid-2000s, values started rising until the 2010s, when the value of the basal diameter reached the level of 1980s and the values of biomass and height reached levels slightly lower than those found in the 1980s. In the case of the Ligurian Sea, we observe a decrease of the values of the parameters until the mid-1990s-2000s with a later increase, in which biomass and basal diamenter values surpassed the values found in the 1960s, as well as some of the height measures found in the 1960s.

In 2009, the CR biomass from the Ligurian Sea reached 1401 g/m$^2$, a value double that reported in 1964 (749 g/m$^2$). The CR biomass showed its lowest value in 1999: 433 g/m$^2$. In the Catalan Sea, biomass ranges from 1269 g/m$^2$ in 1978 to an order of magnitude less, 113 g/m$^2$ in 1992. From the 1990s, CR biomass levels in the Catalan Sea rise again reaching 503 g/m$^2$ in 2013, although the value remains lower than the 1980s value: 636 g/m$^2$ (Fig 4A).

In the studies of the Ligurian Sea, the values reported for basal diameter decrease from 5 mm in 1964 to 2.5 mm in 1994, then increasing and reaching a peak in 2008 (9.32 mm). The most recent measure for basal diameter in the region dates from 2012 and is 6.53 mm, higher than the values reported for the 1960s. However, with the exception of the value from 2008, all other values for basal diameter remain below 7 mm. In the Catalan Sea, basal diameter values decreased from 9.32 mm in 1978 to 2.92 mm in 1991 and then increased to the peak of 7.51 mm in 2011, thus reaching values similar to those found in the late-1970s1980s, with the exception of the peak measured in 1978 (Fig 4B).

The oldest value for CR height from the Ligurian Sea dates from the $3^{rd}$ century BP [43], when its reported value was 15 cm. CR height values in the Ligurian Sea from the 1960s are very variable (1962: 15 cm, 1964: 5.64, 4.88, and 4.77 cm), although there is an overall

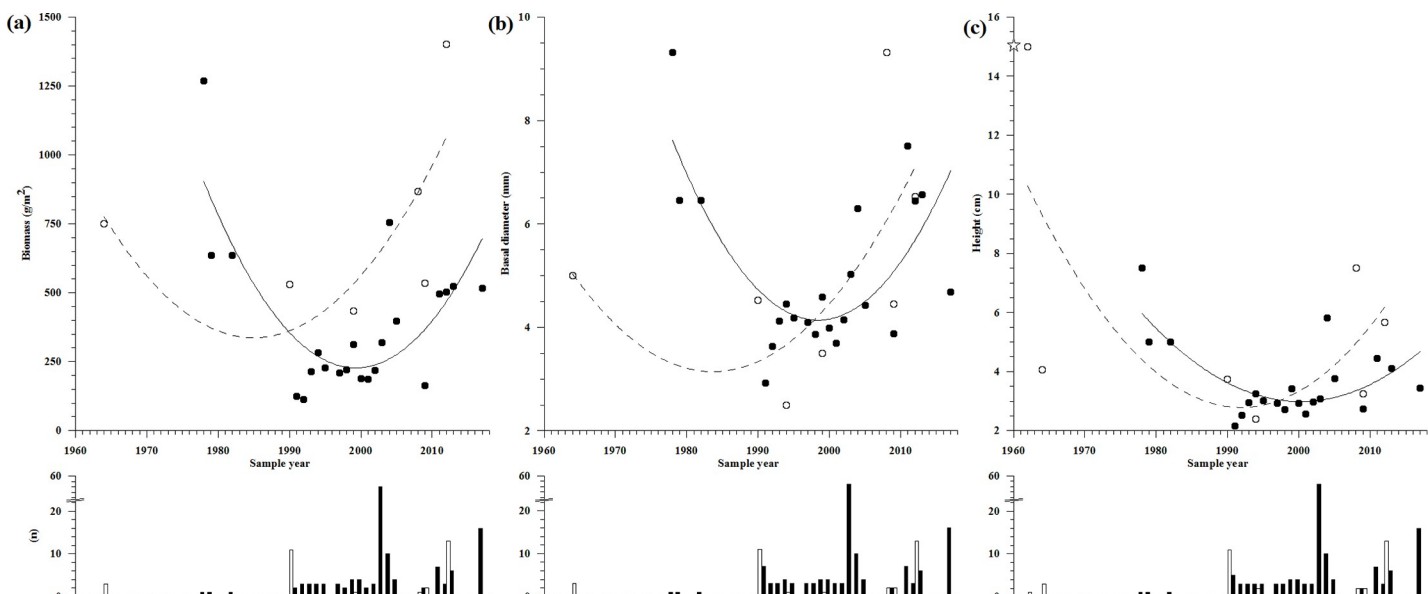

**Fig 4. *Corallium rubrum* parameters by year sampled.** (a) Biomass, (b) basal diameter, and (c) height. Dots correspond to the yearly average value of the parameter. The number of data points used for the year averages are shown below each parameter graph. A data point consists in the value of the parameter from a colony patch (or the closest higher-resolution if not available). Catalan Sea: black dots and bars and continuous curve. Ligurian Sea: white dots and bars and discontinuous curve. The symbol (☆) placed on the Y axis of height corresponds to the year -300 BP for the Ligurian Sea.

decreasing trend, with a minimum value of 2.38 cm in 1994, and then a rise until the peak of 2008: 7.5 cm. Again, the most recent value is from 2012: 5.67 cm, higher than some of the 1960s values. In the Catalan Sea, CR height also decreases from 7.5 cm in 1978 to 2.16 cm in 1991. CR height value then increased to a peak of 4.44 cm in 2011. During the last decade, CR height values were larger than the values reported for the 1990s (2.16–3.41 cm), but smaller than the values reported at the end of 1970s1980s (Fig 4C).

## Carbon sequestration capacity

The three MPAs in the Catalan Sea contain 173.34 ha potentially suitable for CR, which would represent $1.99 \cdot 10^8$ colonies. The estimated carbon sink (carbon invested for growth) of those colonies is 27,419 g C $y^{-1}$, corresponding to 158 g C $ha^{-1}$ $y^{-1}$. For 2017, we estimate $8.94 \cdot 10^5$ kg of CR with a carbon flux of 0.0135 kg C $ha^{-1}$ $y^{-1}$ (C flux = C ingestion–C respiration). The oldest data available for the region, corresponding to 1978, estimate $2.20 \cdot 10^6$ kg of CR and 0.0129 kg C $ha^{-1}$ $y^{-1}$ of carbon flux. The year with the lowest biomass value, 1992, CR would have $1.95 \cdot 10^5$ kg in the region and a flux of 0.0071 kg C $ha^{-1}$ $y^{-1}$.

In the Ligurian Sea, the potential CR occupies 23.8 ha with $5.16 \cdot 10^7$ colonies. The potential colonies C sink accounts for 7,232 g C $y^{-1}$, corresponding to 304 g C $ha^{-1}$ $y^{-1}$. The newest data available from the Ligurian Sea, for 2012, result in the estimation of $3.33 \cdot 10^5$ kg of CR, with a carbon flux of 0.0035 kg C $ha^{-1}$ $y^{-1}$. The oldest data available (1964) and the lowest value (1999) result in $1.78 \cdot 10^5$ and $1.03 \cdot 10^5$ kg of coral, and a carbon flux of 0.0035 and 0.0018 kg C $ha^{-1}$ $y^{-1}$respectively.

## Discussion

Our analyses indicate that variability in CR health status is mainly driven by the harvest intensity and the presence of protection measures. These results, however, should be taken with

caution as data from the last two decades might be biased since they mainly come from protected areas, leaving the recent CR health status in unprotected regions understudied. This bias emphasizes the limitation in unravelling the detailed status of the living communties. Furthermore, the low availability of quantitative pre-1990s data suggest caution in interpreting trends during this time period.

### Biometric parameters analysis

In the Mediterranean region, habitat degradation and exploitation are the main sources of biodiversity loss [65]. These impacts have been occurring since ancient times. For example, coastal modification dates back to the Roman period and coastal exploitation was evidenced by Aristotle (4th Century BP). Over the course of history, the Mediterranean Sea has faced an increasing use of resources, as population living in its coastal areas has grown, maritime trade has intensified, and the commercialization of sea-related products increased [65, 66, 38]. Thus, the shallow water environment where CR can grow has been under pressure, with continuous exploitation since antique periods [13, 21]. Moreover, both chronic and small or intermediate disturbances, as well as sudden large disturbances, have gradually degraded this habitat [67]. In conclusion, CR pristine status before anthropogenic pressure is unknown, as the oldest data available about CR's health status in the Mediterranean Sea was recorded after the human exploitation of the resource had already started, around 30,000 years ago [21, 68]. We can only guess how the "red forests" of the Mediterranean were: possibly large spaces monopolized by CR [69], a slow-growing but very competitive species in terms of space occupation [70].

In this study, the oldest quantitative data available regarding CR status, in the two study regions [71, 72], report an ecosystem already affected by human uses (e.g., trawling, harvesting with devices like the St. Andrew's Cross or the *ingegno*, and/or selective harvesting by divers [9, 13, 21, 64, 73–81]). Some older quantitative information is available for areas not analysed in this study and/or for other health status parameters (e.g. [43, 82, 83]). Data for the regions and the parameters specifically selected in this study are more abundant, and probably more reliable, since the 1990s. Researchers have argued that data on CR harvesting is only reliable from the late-1980s and early-1990s, given that the number of studies prior to these decades is scarce and the oldest techniques to analyse the parameters might be less accurate [84].

Despite the absence of quantitative data, the qualitative data collected in this work suggests that CR was larger, thicker, and had a greater biomass before the 1960s, since when quantitative data became available. Our finding is in line with data from the FAO, which show a sharp decline of 2/3 of the biomass harvested in only few years in the late-1970s [84]. Our result is also consistent with information from the older commercial samples, which usually contain data of the biggest, oldest and less perturbed specimens found in a coral population of a concrete area, as the harvesters seek for the best quality coral, with higher commercial value [64, 76, 81, 85, 86]. Indeed, such samples show the potential status of CR populations if they remain unperturbed from decades to centuries [29]. For the Catalan Sea, a commercial sample collected in 1962, at a depth between 25 to 35 m, and with an average of 16 mm in basal diameter and 11.45 cm in height has been documented while the biggest specimens measured in that sample were 15.5 cm tall, and with a basal diameter of 45 mm [25].

In the 1990s, in the two studied regions, the decreasing trend in CR basal diameter, height and biomass stopped. Several reasons explain this shift: as commercial CR banks get exhausted, the CR fishing decreased relative to previous decades and became stable [86, 87]. The number of harvesting licenses given during this time also decreased [27, 88]. Harvesting of CR was banned in Medes in 1963, and in Cerbère-Banyuls areas where CR cannot be extracted were established in 1974, although they were only effectively implemented after 1981 [89].

Moreover, CR health status parameters compiled in this study rise again in the 2000s, always a bit earlier in the Ligurian Sea compared with the Catalan Sea. However, the CR recovery documented should be interpreted with caution, as it only reflects the community living at a maximum water depth of 60 m in two NW Mediterranean regions. But most importantly, the majority of the data reported in this study comes from MPAs, where different levels of protection measures might positively affect the species' health status. In Portofino, where all the data of Liguria comes from, CR harvest was banned in 1999 [90]; in Medes islands CR cannot be extracted since 1983, a ban that was then extended to the Montgrí Coast in 1992; Cap de Creus has CR protected since 1998 for the partial and integral protected areas [91]; in the Cerbère-Banyuls MPA the CR extraction is also banned since 1979 [92]. Therefore, the results presented here largely illustrate the health status of CR in areas protected since approximately two decades ago, but they cannot be extended to non-protected areas.

However, despite protection, there is evidence that CR poaching has continued regularly in the Catalan Sea [15, 27, 50–52]. Poaching in Liguria seems more sporadic [33], and the harvesting pre-protection less intense [49]. This seems logical as the Catalan Sea has CR of higher quality for the jewellery industry, while that from Liguria is substantially affected by boring sponges [45, 64, 76, 79, 93–96]. In sum, the poaching history and the coral quality might explain why the biometric parameters of the Ligurian CR reached higher values than the ones documented in the 1960s. Probably, the CR status reported in Portofino in the 2010s is similar to how it was in the 1950s, just when intensive harvesting by scuba diving was about to increase [33].

The literature suggests that CR from unprotected areas shows worse health status than the CR in protected areas [15, 91, 92]. Despite such evidence, quantitative studies in non-protected areas still remain scarce and are focused in a few places, although CR presence is known in several other locations such as Llançà [75], l'Escala [97], Llafranch [98], Illes Formigues [64], Blanes, Ametlla de Mar [99], Genoa Bay [71, 100], and La Spezia [100].

Even though available data is not enough to provide a full picture of the health status of CR found deeper than 60 m, we can still hint at its actual ecological trend. The pioneering studies with quantitative data from deep CR are García-Rodríguez & Massó in 1984 [44] and Cattaneo-Vietti et al. in 1994 [45], in the Catalan and Ligurian Seas respectively, both reaching up to 90 m depth. As the study in the Catalan Sea came from a commercial sample, we compare its results with those from a study in the same area in 2002–2003 [14], only counting the specimens $\geq$ 7 mm of basal diameter and $\geq$ 6 cm of height of the 2003 sample. A basal diameter of 13.7 (1984) and 8.07, 10.31, and 10.34 mm (2003), and a height of 11.86 (1984) vs 8.11, 8.48, and 7.28 cm (2003) were recorded. The study of Liguria described samples of 7–9 mm basal width and 10–15 cm tall, meanwhile 2012 samples in the same location had 8, 3, and 6 mm basal diameter and 7, 2.5, 3.2, and 5.1 cm of height [47]. These data, while scarce, seem to suggest that deep corals have been suffering deterioration in recent decades. Such regression is understandable as scuba diving technology improved, allowing deeper harvesting [13, 34, 64, 84], and deep corals remain unprotected by law. Deep coral banks have been historically affected by trawling and trammel nets, but pristine status coral might still exist. Mazzarelli (1915) found CR up to 19 cm tall and 22 mm of basal diameter width in the deep coral banks off Sardinia in 1913 [82]. How close to the pristine size those measurements are will remain unknown.

## Carbon sequestration capacity

Studies like the one presented here are essential to highlight the importance of marine carbon sinks, especially those represented by the animal forests of the seas [4]. The destruction of the

living 3D structures in the benthos diminishes their carbon sequestration capacity by the lack of sessile organisms capable to store it; this implies a loss of mitigation capacity against climate change [37, 61, 101–103], which in turn will wreak future havoc on these habitats [102, 104–106]. A drastic change in some coastal communities may be part of the positive feedback resulting in climate change acceleration, organic pollution, or biodiversity loss [35]. CR has proved to be vulnerable to climate change and ocean acidification. For example, in shallow areas (0–40 m depth), CR has been affected by heat waves [107–112], and can be potentially affected by ocean acidification, which affects the capacity to build its skeleton [113, 114].

In the present study, CR carbon retention capacity was estimated based on the CR data available at different time periods from each study area. Calculations were done assuming that CR extension coincides with the potential habitats it can grow from the locations where practically all the data comes from (Cap de Creus, Medes & Montgrí, Cebère-Banyuls, and Portofino). Furthermore, we assumed a fixed density value despite the fact that density can change in space (location and depth), time, and due to its relation with colony size (self-thinning process) [30, 32]. According to our calculations, since the 1990s until today, the potential carbon retention capacity of CR has nearly doubled. Larger colonies have a larger gonadal output [9, 14, 22], a higher potential recruitment and a higher population stability in the face of perturbations [49].

However, as mentioned, our results are biased towards protected areas, and are hardly transferable to a wider context. For example, a similar calculation of the carbon retained by *Posidonia oceanica* shows a 62 to 87% decrease with respect to data obtained before the 1960s [37], a scenario that might reflect better the reality of the Mediterranean marine ecosystems. Moreover, for our calculations, we decided to be conservative and only consider the CR diameter growth of its base (i.e. the sequestered carbon in the growing rings of its main trunk), based on the mean values observed in both areas [49]. CR branches grow much faster (linear growth 1.5–2.5 mm $y^{-1}$ [115]), for which the inclusion of the carbon stored in each branch in our calculations might have resulted in a higher carbon storage result. If we consider that an ancient red coral may had dozens of branches per colony, the number of sequestered carbon may be exponentially higher compared to the present results.

Results from this work might suggest that CR experienced a dramatic decrease in a short time period until the 1990s, when its carbon retention capacity reached its lowest point. The potential coral weight and the carbon flux decreased nearly two-fold between the oldest and the lowest biomass values separated only by 14 and 35 years in the Catalan and Ligurian Seas, respectively. Such statement needs to be treated with caution due to the low availability of pre-1990s data. CR carbon storage capacity is destrupted by the reduction of its distribution, size and complexity (corals in the past having more branches and polyps and therefore larger capacity to capture carbon and retain it as a structure). An example illustrates the importance of CR distribution for carbon storage: the potential area where CR can grow in the entire Ligurian Sea accounts for 130.9 ha, whereas the actual CR specimens have an extension of 6.31 ha, only found in Portofino MPA [57]. This would account for a C flux of 0.0009 kg C $ha^{-1}$ $y^{-1}$ in 2012, notably smaller than the value extrapolated for the entire Ligurian Sea's potential area (0.0190 kg C $ha^{-1}$ $y^{-1}$).

These kinds of estimates of carbon flux and carbon sink are available in the literature for different benthic systems, but they are still scarce for CR. For example, estimates for seagrass meadows carbon sink gives 6,700 kg C $ha^{-1}$ $y^{-1}$ [103], for *Paramuricea clavata* and *Eunicella singularis* 2,000 kg C $ha^{-1}$ $y^{-1}$ both, and for *Leptogorgia sarmentosa* 0.008 kg C $ha^{-1}$ $y^{-1}$ [116], all of them substantially higher than our results (158 and 304 g C $ha^{-1}$ $y^{-1}$). Again, the difference might be due to the fact that we only considered basal diameter growth. Regarding the C flux, Coppari et al. (2019) [116] calculated 1,000 kg C $ha^{-1}$ for *P. clavata* and *E. singularis*, and 0.02

kg C ha$^{-1}$ for *L. sarmentosa*, which is the closest species to CR in C flux magnitude (0.0135 and 0.0035 kg C ha$^{-1}$ in this study).

## Conclusions

This article provides a spatial and temporal assessment of CR's health status, presenting new understanding of past changes and its carbon sequestration capacity. The CR biometric parameters, and therefore, the species' health status is conditioned by its harvesting pressure and protection measures that impacted its ecological history. In only two to three decades after the establishment of the protected areas, CR reached levels of health similar to those documented in the 1960s (Ligurian Sea) or 1980s (Catalan Sea), suggesting that this protection measures are effective. However, the majority of CR locations remain unprotected and unstudied, leaving no evidence of what would be the actual health status of such colonies. Therefore, to be able to calculate Mediterranean-wide estimates, more quantitative studies about CR demographic and biometric parameters are needed, especially in unstudied locations where CR did not benefit from protective measures. Regarding older CR biometric information, the French Mediterranean coast has an important amount of pre-1990s data (e.g., [83, 117–123]) with several pioneering studies on coralligenous systems (e.g., [124–131]). Meta-analyses of these data could add to the CR trends obtained in this study and shed further light on its ecological history.

The CR size and abundance conditionates its carbon sequestration capacity. Our results suggest that both characteristics can be reduced within a few decades. Further studies on the capacity of carbon retention of the different species present in the coralligenous and other habitats (continental platform, deep water corals . . .) will help to assess the degree of importance of this Mediterranean habitat in the carbon cycle and its climate change mitigation capacity. A more accurate estimation of CR retention (considering the increase of total height and branch lengths) is urgently needed to understand the importance of the marine animal forests as carbon sinks. The knowledge of their longer ecological trends also related to climate change mitigation and biodiversity conservation should improve a decision-making strategy for the species conservation.

## Supporting information

**S1 Fig. In grey the number of documents published per decade with original demographic and biometric information of *Corallium rubrum* (CR) within the Catalan Sea (including Côte Vermeille), Ligurian Sea, and off Tuscany regions: n = 84.** The documents that provide qualitative information represent 29% of the total. In cross grids the proportion of documents containing original data about CR basal diameter, height and/or weight: n = 37.
(PDF)

**S2 Fig. *Corallium rubrum* density values by year sampled.** Dots correspond to the yearly average value of density. The number of data points used for the year averages are shown below. Catalan Sea: black dots and bars. Ligurian Sea: white dots and bars.
(PDF)

**S1 Table. List of publications of *Corallium rubrum* with demographic, biometric, and location information from the Catalan Sea, Ligurian Sea and off Tuscany.** The table shows the main characteristics and parameters described of the publications included in the meta-analysis, plus the publications excluded. The publications come from an extensive literature review in the three regions studied (further information on S1 Text).
(PDF)

**S2 Table. Qualitative information related with the health status of *Corallium rubrum* from the Catalan Sea, Ligurian Sea and off Tuscany.** The information comprises notes or text extractions from the original documents. For the full reference of each code in the first column see the list of references of S1 Table.
(PDF)

**S3 Table. PRISMA checklist.** Identification of the main items of the meta-analysis of this study and the page number where they are reported.
(PDF)

**S4 Table. *Corallium rubrum* values of its most studied parameters and number of colonies sampled for obtaining them.** These values are used as data points for the yearly averages in Fig 4. Weight values are multiplied by a fixed density value to show biomass values in Fig 4.
(PDF)

**S1 Text. Supporting information methodology.** This article was elaborated through a simplification of an original literature review. The methodology of such review is explained here, as well as the source of Fig 3, S1 and S2 Tables, plus S1 Fig.
(PDF)

**S1 Appendix. Data underlying figures and calculations.**
(XLSX)

## Acknowledgments

We thank the Biblioteca Carles Bas i Peired (CSIC-CMIMA, Barcelona) and the Bibliothèque du Laboratoire Arago (BUPMC, Banyuls-sur-mer) for facilitating an essential part of the literature review of this study. In special, to Ignacio J. Castaño Pacho (CSIC-CMIMA) and Sandrine Bodin (BUPMC) for their guide and help during the literature research. Additional thanks to L. Bramanti for permit, access and accommodation facilities and to P. Graham Mortyn for the English proofreading of this article. We also thank R. Cattaneo-Vietti and J. Evans for their help to improve the manuscript. The authors thank the Generalitat de Catalunya (MERS) for their support (2017 SGR—1588). This work contributes to the ICTA-UAB "Unit of Excellence" (MDM2015-0552). Finally, thanks to all the scientists that help us to obtain rare documents.

## Author Contributions

**Conceptualization:** Miguel Mallo, Patrizia Ziveri, Victoria Reyes-García, Sergio Rossi.

**Data curation:** Miguel Mallo.

**Formal analysis:** Miguel Mallo, Sergio Rossi.

**Methodology:** Miguel Mallo, Patrizia Ziveri, Victoria Reyes-García, Sergio Rossi.

**Supervision:** Patrizia Ziveri, Victoria Reyes-García, Sergio Rossi.

**Validation:** Patrizia Ziveri, Victoria Reyes-García, Sergio Rossi.

**Writing – original draft:** Miguel Mallo.

**Writing – review & editing:** Miguel Mallo, Patrizia Ziveri, Victoria Reyes-García, Sergio Rossi.

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
