## [Decision Letter · Decision Letter 0]

16 Jul 2019

PONE-D-19-15450

Historical record of Corallium rubrum and its changing carbon sequestration capacity: a meta-analysis from the North Western Mediterranean

PLOS ONE

Dear Mr. Mallo,

Thank you for submitting your manuscript to PLOS ONE. After careful consideration, we feel that it has merit but does not fully meet PLOS ONE’s publication criteria as it currently stands. Therefore, we invite you to submit a revised version of the manuscript that addresses the points raised during the review process.

Both reviewers suggested minor changes, which the authors can easily introduce in their revised ms. One of the reviewers evidenced some methodological issues, and I recommended the authors to take full consideration of them.

We would appreciate receiving your revised manuscript by Aug 30 2019 11:59PM. To enhance the reproducibility of your results, we recommend that if applicable you deposit your laboratory protocols in protocols.io, where a protocol can be assigned its own identifier (DOI) such that it can be cited independently in the future. For instructions see: http://journals.plos.org/plosone/s/submission-guidelines#loc-laboratory-protocols

We look forward to receiving your revised manuscript.

Kind regards,

Carlo Nike Bianchi

Academic Editor

PLOS ONE

Journal Requirements:

Reviewers' comments:

Reviewer's Responses to Questions

**Comments to the Author**

1. Is the manuscript technically sound, and do the data support the conclusions?

Reviewer #1: Yes

Reviewer #2: Partly

2. Has the statistical analysis been performed appropriately and rigorously? 

Reviewer #1: Yes

Reviewer #2: Yes

3. Have the authors made all data underlying the findings in their manuscript fully available?

Reviewer #1: Yes

Reviewer #2: No

4. Is the manuscript presented in an intelligible fashion and written in standard English?

Reviewer #1: Yes

Reviewer #2: Yes

5. Review Comments to the Author

Reviewer #1: Manuscript Draft: PONE-D-19-15450

Historical record of Corallium rubrum and its changing carbon sequestration capacity: a meta-analysis from the North Western Mediterranean

The Authors described and related the Corallium rubrum health status in the North Western Mediterranean, in terms of morphometric variables and, potentially, as carbon sink species. A strong effort has been done to collect an important and complex data collection.

This has permitted to reach some conclusions. Obviously the different variables (basal diameter, height and / or weight for colony) were related to each other. The result more interesting is the recovery, in terms of colony size, of the protected populations after the heavy harvesting occurred in the last century. This trend, already known for some sites, today seems common to all the sites explored by the authors, suggesting that the protection measures have been effective. And this is an excellent result.

Sampling methods and statistics were performed correctly and described in detail.

In my opinion, the results are interesting, and the paper can be accepted with minor revision.

Same small remarks were done directly on the ms.

Reviewer #2: General remarks:

Mallo et al. have collated existing data on Corallium rubrum colony mormphometric parameters from the Catalan and Ligurian Seas in order to elucidate temporal trends in these parameters, and also estimate carbon sequestration rates. This is an interesting work but there are some aspects of the methodology that need attention, or at least better justification.

Biometric parameters:

As with any work that relies on pre-existing data, a major limitation (that the authors have no control over) is the limited availability of historical data. Although the authors highlight this issue, I believe that they have not sufficiently shown whether the available data, particularly for pre-1990, is still adequate in order to make inferences on trends or otherwise. In particular, in Figure 4, the number of data points used for the year averages is exceedingly low (again, especially before 1990). For Liguria, the entire U-shaped pattern is reliant on the yearly average for 1964, which is the only pre-1990 value included and is based solely on three data points. For Catalan Sea, there are only three pre-1990 values in the graphs, each based on a single data point only. I strongly recommend that the authors: (i) explain better what a ‘data point’ is; (ii) give an indication of the number of colonies on which the yearly average data in Figure 4 is based, for each year; (iii) give due consideration to the lack of existing data when interpreting their findings, clearly indicating which of the observed patterns may be artefacts of low data availability.

The authors do not appear to take spatial or depth variations into account in any way in their study. Could some of the differences in the yearly averages reported be simply due to the data being collected from different stations or locations? In addition, the authors rightly highlight that recent data mostly originates from studies in MPAs and may not be representative of the situation elsewhere. Is the data for pre-1990 also from the same areas that are now MPAs?

Carbon sequestration:

The statement [L216] that “The potential area where CR can live was estimated from the literature” requires further explanation. What is meant by ‘potential area’, and how was this estimated? In my opinion it is not enough to state that this was done based on data from MPAs. Is it the total area within the mentioned MPAs where Corallium rubrum is known to occur? Or where the habitat is considered suitable for Corallium rubrum, even if it is not present? Is the data from MPAs being used to extrapolate to get an estimate of ‘potentia; area’ that includes the outside regions too? If so, what are the boundaries for this area estimation? From the discussion L525-531], it appears that a distinction is being made between the ‘potential area’ and ‘actual area’, with the ‘potential area’ extending beyond the MPA boundaries. This is really not obvious from the Methods section.

I consider the use of a fixed density value for estimating total coral biomass [L208-215] as problematic for two reasons. First, the very existence of widely varying density data (S2 Fig), while possibly due to different methodologies, is also likely a result of actual spatial variation in coral density (e.g. Scandola has higher Corallium rubrum densities than elsewhere, as highlighted in L531-537). If density varies from site to site, then using a fixed density value may lead to an inaccurate estimate of total carbon sequestration. More problematic, however, is that this fixed density value is being combined with the colony size measurements to estimate carbon sequestration rates. But with density fixed, the carbon sequestration rate becomes a function of colony size. Yet Cau et al. (2016) have strongly indicated that Corallium rubrum populations are influenced by self-thinning processes, where population density and mean colony size are inversely related. Therefore, it is not appropriate to assume that density is fixed and that only colony size is changing.

Other general remarks:

The PLOS Data policy requires authors to make all data underlying the findings described in their manuscript fully available without restriction… For example, in addition to summary statistics, the data points behind means, medians and variance measures should be available.

The authors have only partly addressed this requirement, because the actual data used to calculate the yearly average values shown in Fig. 4 is not provided anywhere. The authors should include these data as supplementary material, either as part of S1 Table or as an additional supplementary table.

Specific comments:

L200-207: Higher r² statistically means that most of the variation in ‘Y’ is explained by ‘X,’ and therefore that knowing the ‘X’ value one should be able to get a good prediction on the ‘Y’ value. It is therefore very strange to see that Eqn 3, which had the highest r² value, is indicated as having the greatest prediction error, while Eqn 2, which had the lowest r², is shown as having the smallest error.

L420-423: The authors are attributing changes in biometric parameters from the 1990s onward to the to the ban of trawling and ingegno fishing. But my impression is that colonies found at the depths covered by the present work, i.e. down to 60m, were especially targeted by Scuba diving, whereas trawling and ingegno fishery was used more at deeper depths.

Several other minor comments or suggestions are included in an annotated version of the manuscript.

Reference:

Cau et al (2016) Habitat constraints and self-thinning shape Mediterranean red coral deep population structure: implications for conservation practice. Scientific Reports, 6:23322. DOI: 10.1038/srep23322

6. PLOS authors have the option to publish the peer review history of their article (what does this mean?). If published, this will include your full peer review and any attached files.

Reviewer #1: Yes: Riccardo Cattaneo-Vietti

Reviewer #2: No

---

## [Author Response · Author response to Decision Letter 0]

30 Aug 2019

You can find the specific answers to the reviewers comments in the document attached "Response To Reviewers".

---

## [Decision Letter · Decision Letter 1]

13 Sep 2019

PONE-D-19-15450R1

Historical record of Corallium rubrum and its changing carbon sequestration capacity: a meta-analysis from the North Western Mediterranean

PLOS ONE

Dear Mr. Mallo,

Thank you for submitting your manuscript to PLOS ONE. After careful consideration, we feel that it has merit but does not fully meet PLOS ONE’s publication criteria as it currently stands. Therefore, we invite you to submit a revised version of the manuscript that addresses the points raised during the review process.

One of the reviewers spotted some further errors in the revised manuscritp and asked for very minor changes that the authors can easily introduce.

We would appreciate receiving your revised manuscript by Oct 28 2019 11:59PM. To enhance the reproducibility of your results, we recommend that if applicable you deposit your laboratory protocols in protocols.io, where a protocol can be assigned its own identifier (DOI) such that it can be cited independently in the future. For instructions see: http://journals.plos.org/plosone/s/submission-guidelines#loc-laboratory-protocols

We look forward to receiving your revised manuscript.

Kind regards,

Carlo Nike Bianchi

Academic Editor

PLOS ONE

Reviewers' comments:

Reviewer's Responses to Questions

**Comments to the Author**

1. If the authors have adequately addressed your comments raised in a previous round of review and you feel that this manuscript is now acceptable for publication, you may indicate that here to bypass the “Comments to the Author” section, enter your conflict of interest statement in the “Confidential to Editor” section, and submit your "Accept" recommendation.

Reviewer #1: All comments have been addressed

Reviewer #2: (No Response)

2. Is the manuscript technically sound, and do the data support the conclusions?

Reviewer #1: Yes

Reviewer #2: Yes

3. Has the statistical analysis been performed appropriately and rigorously? 

Reviewer #1: Yes

Reviewer #2: Yes

4. Have the authors made all data underlying the findings in their manuscript fully available?

Reviewer #1: Yes

Reviewer #2: Yes

5. Is the manuscript presented in an intelligible fashion and written in standard English?

Reviewer #1: Yes

Reviewer #2: Yes

6. Review Comments to the Author

Reviewer #1: The manuscript describe a technically sound piece of scientific research with data that supports the conclusions. The conclusions are drawn appropriately based on the data presented. I suggest the publication, being the revision suggestions accepted

Reviewer #2: I have read with interest the revised version of the manuscript by Mallo et al. on historical records of Corallium rubrum and consequent changes its carbon sequestration rate, as well as their detailed response letter to the reviewers’ feedback.

I consider that the authors have adequately addressed my previous comments and the manuscript is now nearly acceptable for publication, but recommend the following minor amendments:

L53-55: “The halt in the C. rubrum decreasing trend coincided with the status of overexploitation in the existing populations” >> The meaning of this phrase is not fully clear to me… Perhaps: “The halt in the C. rubrum decreasing trend coincided with overexploitation of the existing populations”?

L38: “of” should not be in italics

L73: “Moreover, compared to larger colonies, the younger (and smaller) colonies possibly retain less carbon in their structures” >> In their reply, the authors have indicated that “We refer here to benthic suspension feeders that retain carbon. Apart of anthozoans we might find also bryozoans, polychaetes, sponges…” Since polychaetes and sponges are not colonial, the term “colonies” may not be the best choice of word, so I suggest they consider alternative wording.

L146: “antiguity” should be “antiquity”

L165: It would be ideal to briefly what “colony patch” means…

L166-168: “The average of all the colony patches reported parameters (or highest resolution) sampled in the same year, conformed our yearly mean value.” >> I suggest rewording this sentence to: “For a given parameter, the yearly mean value was taken as the average of the values for all the colony patches (or higher resolution) sampled in the same year.”

L226: “From the Ligurian Sea” should be “For the Ligurian Sea”

L508: “Furthermore, we assumed a fixed density value despite density could change in space (location and depth), time, and due to its relation with the colony size (self-thinning process)” is better written “Furthermore, we assumed a fixed density value despite the fact that density can change in space (location and depth), time, and due to its relation with colony size (self-thinning process)”

7. PLOS authors have the option to publish the peer review history of their article (what does this mean?). If published, this will include your full peer review and any attached files.

Reviewer #1: No

Reviewer #2: Yes: Julian Evans

---

## [Author Response · Author response to Decision Letter 1]

25 Sep 2019

Response to reviewers can be found attached in an Office Word file.

---

## [Editor Report · Decision Letter 2]

30 Sep 2019

Historical record of Corallium rubrum and its changing carbon sequestration capacity: a meta-analysis from the North Western Mediterranean

PONE-D-19-15450R2

Dear Dr. Mallo,

We are pleased to inform you that your manuscript has been judged scientifically suitable for publication and will be formally accepted for publication once it complies with all outstanding technical requirements.

With kind regards,

Carlo Nike Bianchi

Academic Editor

PLOS ONE
---

## [Editor Report · Acceptance letter]

19 Nov 2019

PONE-D-19-15450R2 

Historical record of *Corallium rubrum* and its changing carbon sequestration capacity: a meta-analysis from the North Western Mediterranean 

Dear Dr. Mallo:

I am pleased to inform you that your manuscript has been deemed suitable for publication in PLOS ONE. Congratulations! Your manuscript is now with our production department. 

With kind regards,

on behalf of

Dr. Carlo Nike Bianchi 

Academic Editor

PLOS ONE